# Knowledge and Awareness of Polish Parents on Vaccination against Human Papillomavirus

**DOI:** 10.3390/vaccines10071156

**Published:** 2022-07-20

**Authors:** Piotr Sypień, Tadeusz M. Zielonka

**Affiliations:** 1Sebastian Petrycy Health Care Facility in Dąbrowa Tarnowska, Szpitalna Street 1, 33-200 Dąbrowa Tarnowska, Poland; piotr.sypien@gmail.com; 2Department of Family Medicine, Medical University of Warsaw, S. Binieckiego Street 6, 02-097 Warsaw, Poland

**Keywords:** human papillomavirus vaccines, sexually transmitted diseases, oncological risk, health education, cancer prevention

## Abstract

**Background**: Human papillomavirus (HPV) vaccinations are rare among Polish children, and the reasons are scant. The objective was to evaluate the knowledge, attitude and awareness of parents about HPV vaccination to investigate reasons for low HPV vaccination coverage. **Methods**: 387 parents of children hospitalized at the Children’s Hospital were asked to participate in an anonymous and voluntary survey study. Three hundred and two surveys were returned. **Results**: Only 54% of participants have heard about HPV, while 26% know that it is a sexually transmitted disease. According to 71% of responders, vaccines are generally effective, and 63% claim that they are safe. However, only 5% of daughters and 4% of sons are vaccinated against HPV. A total of 25% of parents spoke with their doctor about HPV-related diseases and prevention methods. A higher level of education (*p* = 0.01), knowledge of sexually transmitted diseases (*p* < 0.0001), perceiving vaccination as an effective and safe prophylactic method (*p* < 0.0001), and conversations with a doctor (*p* < 0.0001) are strong motivators to vaccinate children against HPV. This decision is free of religious beliefs, origin, age, gender and the number of children. **Conclusions**: Polish parents have a positive attitude towards vaccination. They recognize the limitations of their knowledge and express a desire to further it. Educational activity is an important element of physicians’ work.

## 1. Introduction

Human papillomavirus (HPV) is the most common sexually transmitted infection and is usually asymptomatic [1]. Some oncogenic types of HPV can cause carcinogenesis and lead to cancer of the cervix, vagina, vulva, penis, rectum and oropharynx [2]. Cervical cancer is the fourth most common cancer among women and the second most common in the female population between the ages of 25 and 40 [3]. Every year, approximately 600,000 women worldwide are diagnosed with this cancer [1]. It is also a vital issue in Poland, with 3862 new cases in 2020 and 2137 deaths, and this morbidity is higher than the average for EU countries [4].

Oncological screening is crucial in the early detection of precancerous lesions in cervical cancer and should be performed regularly among Polish women [5]. Adherence to prevention programs is crucial for women, and proper awareness is necessary [6].

The precede-proceed system may be recommended in order to increase the effectiveness of screening tests; causes include various external factors and are assessed for possible modifications, ensuring the effectiveness of the test [7].

The introduction of HPV vaccines has reduced the morbidity and mortality associated with HPV diseases, and long-term observations have confirmed their safety profile [8,9]. Currently, 2-, 4-, and 9-valent vaccines are optional but recommended by the Ministry of Health in Poland for children before sexual initiation, but there is no national immunization program [10]. HPV vaccination coverage in Poland includes around 7% of adolescents compared to 62% in Italy, 75% in Australia and 80% in England [11,12,13,14]. At the same time, vaccination coverage against other infectious diseases in Poland is much higher.

Financial barriers and the lack of a state-organized HPV vaccination program have already been highlighted by parents and doctors as a significant limitation to this vaccine development in Poland [15]. The effective implementation of new strategies that decrease HPV-related diseases requires the proper preparation of the public. Public discussions of HPV vaccines have often been about adolescent sexuality and sex education. Additionally, an increase in the generally negative reception and fear of vaccination has contributed to an increased denial of vaccinations that are already co-financed and covered by immunization programs in Poland. Changes in perceptions of vaccination indicate the need for new studies on attitudes towards vaccination among parents to prepare effective immunization programs. Health behaviors are influenced by multiple factors—educational, psychological, sociological, political, economic, administrative and environmental—which must be considered and assessed for their modifiability.

Why do so many parents in Poland vaccinate their children against many diseases, and why do so few vaccinate them against HPV? It is necessary to know both the reasons that motivate people to vaccinate and the barriers that limit their willingness. The purpose of this study was to assess the knowledge and awareness of Polish parents about HPV vaccination and the general attitude regarding vaccines as a method of prophylaxis to reduce infectious diseases, and to investigate the reasons for the low HPV vaccination coverage in Poland.

## 2. Methods

This anonymous cross-sectional study was conducted among 387 parents of children hospitalized at the University Children’s Hospital in Cracow, Poland. The inclusion criteria were the voluntary completion and return of the questionnaire by parents present in the Hospital on the days of 20–24 September 2021. The exclusion criterion was not agreeing to participate in the study. The study sample consisted of 302 participants (78%, Table 1). The data was collected using a self-administered survey consisting of 32 single- and multiple-choice questions. The questions focused on socio-demographic data, knowledge about a healthy lifestyle and disease prevention methods, sources of health information, general opinions about vaccinations, familiarity with HPV infections and the parents’ willingness to vaccinate their own children against HPV.

The statistical analysis and categorical data were presented as frequencies with percentages. The chi square Pearson test was used to compare two groups of categorical variables. The significance level was assessed as 5% in statistical analyses (*p* < 0.05).

## 3. Results

### 3.1. General Attitude towards Vaccines

According to 71% of participants, vaccines are effective methods for reducing morbidity and mortality from infectious diseases. Additionally, 63% of participants claim that they are safe (Table 2). For sources of information about health and vaccines, parents indicated mainly the Internet (77%), followed by physicians (60%), TV/radio (55%), news outlets (34%), public health campaigns (32%), family or friends (22%) and school (16%). Almost two-thirds of parents expressed a need to increase their knowledge about vaccines. Most of the respondents (80%) declared that their children are vaccinated according to the Polish vaccination schedule, and 25% of them opted for additional vaccinations. When deciding on voluntary vaccination, the type of disease was important to 21% and the cost to 25% of parents. No correlation was found between making the decision about performing a vaccination of their own children and the age (*p* = 0.11), residency (*p* = 0.16), education level (*p* = 0.52) and religious beliefs (*p* = 0.92) of the parents that were questioned.

More than half of responders (54%) favored a free distribution of vaccines, 19% supported the co-funding of mandatory vaccines only, 22% were in favor of sharing the cost of vaccines between the patient and the government, and 5% believed that all vaccines should be fully paid for.

### 3.2. Knowledge and Awareness about Sexually Transmitted Diseases

Only 54% of participants, especially women (59% vs. 40%; *p* = 0.002), had heard of HPV. Regarding the sexually transmitted diseases, 82% of participants indicated that they had heard of HIV/AIDS, 73% syphilis, 65% gonorrhoea, 28% trichomoniasis and hepatitis C, and 26% HPV infection. Moreover, 12% of responders reported being aware of all sexually transmitted diseases, and 34% of them knew of the potential oncologic risks of chronic HPV infection. Only 28% of parents knew that a viral infection cannot be cured with antibiotics. For 83% of parents, medical education at school is insufficient and there is a strong need to pay more attention to healthy lifestyle promotion and disease prevention, including sexually transmitted diseases. Two-thirds of participants reported having conversations about a healthy lifestyle with their own children. However, only 35% spoke with their children about sex, or sexually transmitted diseases and prevention methods. On the other hand, 14% did not feel ready to talk about it. In the opinion of 23% of the questioned parents, HPV vaccination might make adolescents more likely to participate in sex at an earlier age.

### 3.3. Attitude towards HPV Vaccines

As a potential limitation to performing HPV vaccinations, responders listed, first of all, a lack of sufficient knowledge about this vaccine (51%) and a lack of awareness about the need to be vaccinated (34%), followed by fear of side effects and high cost (27%), non-mandatory status (12%), insignificant effectiveness (7%), personal negative attitude to vaccines (7%), negative news about vaccination from media (3%) and lack of doctors’ recommendation (1%). Care about health (33%) and fear of cancer (21%), followed by medical recommendations (19%), local government vaccination actions (14%), self-conviction about positive role of vaccines in prevention of diseases (13%) and positive information materials from media (6%) were rated by participants as sources of motivation to vaccinate their children against HPV.

A quarter of the participants had the opportunity to talk to a doctor about HPV and prevention methods. Medical consultations were led by a pediatrician (39 answers), a gynecologist (6), a general practitioner (4) or other (3). Participants living in a town containing over 100,000 inhabitants have a greater availability to doctor’s information and recommendations (34% vs. 19%; *p* = 0.005); the same applies to those more highly educated (34% vs. 17%; *p* = 0.001). No statistically significant correlations were found regarding responders’ gender (*p* = 0.69) and age (*p* = 0.73). In the opinion of 13% of questioned parents, doctors recommend vaccinating children against HPV during medical check-ups, while 30% claim that physicians spend too little time on this problem and 54% report that physicians do not speak about this topic at all. Only 5% (17 answers) of parents vaccinated their daughters and 4% (14 answers) vaccinated their sons. Additionally, 157 will vaccinate their child: 100 daughters and 57 sons. One hundred and fifty-four responders (45%) did not or will not vaccinate. Positive opinions regarding HPV vaccination expressed as a vaccination or willingness to vaccinate their own child depended on the education level of parents, medical knowledge and conversations with physicians about immunization (Table 3).

## 4. Discussion

An understanding of parental awareness and attitudes is essential to uncover the reasons for the low HPV vaccination coverage and try to increase it, since parents are responsible for promoting healthy lifestyles in their children. This study confirmed that Polish parents perceive vaccines as an effective and safe method for limiting infectious diseases. This opinion is mainly seen among people with a higher education, living in big cities and having many children. However, parents still rarely choose HPV vaccinations. According to our results, unfamiliarity with HPV-related diseases is a main limitation in the decision to get this vaccination. This study showed that parents are aware of the limitations of their knowledge regarding vaccines, which have a clear negative impact on their decision to vaccinate their children. At the same time, they are open to learning more. This is a particularly important factor in creating public health awareness in the community. The Internet is quickly developing as the most important means of communication, providing an opportunity to effectively reach a wide audience. However, it is also a space for movements undermining the logic of vaccinations and evidence-based medicine [16,17]. There is, therefore, a clear need for top-to-bottom health promotion programs in the form of social campaigns, especially in the virtual world, that can effectively familiarize parents with the problem of HPV. Success of media campaigns regarding HIV/AIDS is confirmed by the widespread public awareness of the risks of infection during sexual intercourse, which has been demonstrated in our study and others [18]. The most important source of health information continues to be the traditional mass media [19].

Communication and education strategies must be undertaken to ensure that parents are fully informed [20]. The knowledge about HPV among Polish parents does not differ much from parents in other developed countries [21,22]. Despite the common prevalence of the disease, only one-third of parents are aware of the cancer risk. This is particularly a problem for a country without a national HPV immunization program, where the realization of this vaccination depends on a parental decision. Healthcare workers play an essential role in providing medical information [21,23]. However, according to the participants, doctors rarely discuss this with them and spend too little time on the HPV problem. Parents’ opinions challenge doctors’ statements about the frequency of their conversations with patients about HPV [24]. Adequate physician-patient communication is an essential source of increased public awareness of disease and prevention methods. Healthcare providers should be encouraged to conduct educational activities, and the organization of their work time should include more preventative coverage, because this study demonstrated that parents who talk to doctors about HPV are significantly more likely to have a pro-protective attitude towards their children. We also showed that people living in big cities have better access to more educated doctors who are able to share knowledge. This is a social problem, related to inequalities in the access to skilled medical care. The limited availability of medical care to rural residents has also been recognized in other countries [25]. In the United States, a variety of methods are proposed to level the playing field, for example through expanded funding for vaccination programs, school entry requirements, and primary healthcare practices instituting orders for the first dose of HPV vaccination during the medical visit and subsequent doses in a local pharmacy [26].

The decision to vaccinate a child against HPV is not an easy one for Polish parents, and very few children were vaccinated. However, many responders declared their willingness to get their children vaccinated, especially regarding girls, which is also noted in other countries [27,28]. The motivating factors for vaccination vary. Knowledge of HPV-related diseases and discussions with a doctor about it are especially important factors in building social awareness. Parents’ opinion on vaccination is free from neither restrictions in their background nor gender. Freedom from religious beliefs should be seen as a positive characteristic of Polish parents (despite the various discussions about religious affiliation, including politicians’ arguments against vaccine reimbursement), and it also compares favorably to other countries [29]. Another problem that was shown in this study was the public’s limited awareness about the treatment of viral infections. Over two-thirds of participants answered that they believed HPV infection could be treated with antibiotics. Such a statement was also pointed out in other studies [30,31]. The misconception about antibiotic protection against viral diseases can lower parents’ vigilance against infections and cast doubt on the wisdom of vaccination.

Participants negatively evaluate schooling in terms of preparing their children for disease prevention. For this reason, parent-child conversations about healthy lifestyles are an important part of the education and prevention of risky sexual behavior [32,33]. The majority of parents surveyed said they talk to their children, but a significant percentage do not talk or are not prepared to talk about sexual education with their children. This is an indication that attempts to prepare parents for such conversations are needed.

The study also has some limitations. It was conducted on a small sample of people in one region of the country. However, it was a randomly selected group comprising all parents who spent five days (20–24 September 2021) with their sick children in one of the hospitals in Krakow. Of this group of 387 parents, 78 percent took part in the survey. Survey participants declared their own responses, so there is a risk of providing unreliable information. Due to the fact that the survey was conducted in a highly specialized hospital in Poland, many children are hospitalized with chronic diseases and thus have more contact with medical providers and are more interested in health. Therefore, their answers may be better than those in the general population.

## 5. Conclusions

Polish parents perceive immunization as an effective and a safe form of prevention from infectious diseases, and they vaccinate their children willingly. However, barriers to vaccination are perceived to be due to the cost of additional vaccinations in the Polish healthcare system and participants’ declared lack of adequate knowledge and awareness of the need to vaccinate.

The research confirms that the childhood HPV vaccination rates are still low; however, a significant proportion of respondents expressed their willingness to vaccinate. This was independent of religious beliefs, age and gender. Those who were more open to vaccination were parents with a higher education, living in big cities and having families with many children.

Participants are aware of their ignorance and are willing to correct it. They mainly rely on media and Internet content, which is not always verified. However, medical advice is also an important factor in knowledge acquisition. Unfortunately, doctors talk about prevention with their patients far too rarely. There should be more focus on creating pro-health attitudes and prevention with the help of healthcare providers, as they inspire public trust and provide knowledge in line with evidence-based medicine.

Also indicated by parents is the importance of a high-quality school education and its role in creating appropriate attitudes towards public health. The participants themselves stressed that they had difficulties in communicating with their children, especially when it came to difficult topics such as sexual education and pro-health attitudes.

## Figures and Tables

**Table 1 vaccines-10-01156-t001:** Characteristics of the population study.

Parameter	Feature	Number	%
** *Gender* **	Female	212	70
	Male	90	30
** *Origin* **	Rural area	77	25
	Town with less than 20,000 habitants	71	24
	Town between 20,000 and 100,000 habitants	42	14
	Town between 100,000 and 500,000 habitants	57	19
	Town over 500,000 habitants	54	18
** *Education level* **	Basic	15	5
	Secondary	55	18
	Vocational	100	34
	Advanced	132	44
** *Religious beliefs* **	Yes	248	82
	No	54	18

**Table 2 vaccines-10-01156-t002:** General participants’ attitude towards vaccines.

			Vaccinations as An Effective Method of Reducing Infectious Diseases (*n* = 214)			Do you Consider Vaccines as A Safe Method of Prevention? (*n* = 192)		
Participants (*n* = 302)		%	Number	%	*p*	Number	%	*p*
** *Gender* **								
female	212	70	143	67	0.05	128	60	0.08
male	90	30	71	79		64	71	
** *Age* **								
≤34	164	54	113	69	0.41	98	60	0.13
>34	138	46	101	73		94	68	
** *Origin* **								
Rural area or town ≤ 100,000 hbs	190	63	126	66	**0.02**	111	58	**0.02**
Town > 100,000 hbs	112	37	88	78		81	72	
** *Education level* **								
Low or medium	170	56	100	59	**0.001**	83	49	**0.001**
High	132	44	114	87		109	83	
** *Number of children* **								
≤2	248	82	187	75	**0.0002**	164	66	**0.05**
>2	54	18	27	50		28	52	
** *Religious beliefs* **								
Yes	248	82	177	71	0.68	156	63	0.60
No	54	18	37	69		36	67	

**Table 3 vaccines-10-01156-t003:** Positive attitude towards HPV immunization of own child.

	Participants (*n* = 302)			Positive Attitude toward Vaccination of Own Child (*n* = 147)		
	Answers	%		Answers	%	*p*
** *Gender* **						
female	212	70		47	99	0.29
male	90	30		48	53	
** *Age* **						
≤34	164	54		85	52	0.23
>34	138	46		62	45	
** *Residency* **						
rural area or town ≤ 100,000 hbs	190	63		88	46	0.29
town > 100,000 hbs	112	37		59	53	
** *Education level* **						
low or medium	170	56		61	36	**0.001**
high	132	44		86	65	
** *Number of children* **						
≤2	248	82		125	50	0.2
>2	54	18		22	41	
** *Religious beliefs* **						
yes	248	82		121	49	0.93
no	54	18		26	48	
** *Perceive vaccinations as an effective method of reducing infectious diseases* **						
yes	214	71		132	62	**<0.0001**
no/I don’t know	88	29		15	17	
** *Perceive vaccines as a safe* **						
yes	192	64		117	61	**<0.0001**
no/I don’t know	110	36		30	27	
** *Tumor history in family* **						
yes	167	55		86	52	0.28
no	135	45		61	45	
** *Cognizance of sexually transmitted disease* **						
yes	35	12		32	91	**<0.0001**
no	267	88		115	43	
** *HPV infection can be treated with the use of antibiotics* **						
yes	216	72		90	42	**0.0002**
no	86	28		57	66	
** *Spoke with doctor about vaccination* **						
yes	227	75		84	37	**<0.0001**
no	75	25		63	84	
** *Reimbursement would encourage vaccination* **						
yes	196	65		113	58	**0.0002**
No	106	35		34	32	

## Data Availability

Data is available after contacting the corresponding author.

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
