# Peer review of "Knowledge and Awareness of Polish Parents on Vaccination against Human Papillomavirus"

_vaccines, 2022, doi:10.3390/vaccines10071156_

Round 1
Reviewer 1 Report
Dear Editor-in-Chief
Manucript : ¨Knowledge and awareness of Polish parents on vaccination against human papillomavirus¨ by Piotr Sypień, Tadeusz M Zielonka
The quality of this manuscript in low and these conclusions can not be supported by the low size samples.
The interest of this study is low for a general audience in Vaccines (MDPI). In addition, authors have not indicates whihc statistical method they followed for p stimation in material and methods. The conclusion is poor and it is not really supported by these finginds given the low size sample.
Thanks¡
Author Response
The quality of this manuscript in low and these conclusions can not be supported by the low size samples.
I agree that the number of people taking part is not large. However, it was a randomly selected group of all parents who spent five days (September 20-24) with their sick children in one of the hospitals in Krakow. Of this group of 387 parents, 78 percent took part in the survey. A limitation was the fact that these were people who had sick children who required hospitalization. It is worth comparing this with a group of parents who have healthy children. We have added a comment about these limitations in the discussion.
The interest of this study is low for a general audience in Vaccines (MDPI).
This article was submitted for a special issue on the sociological aspects of vaccination. Therefore, it is different from the typical topic.
In addition, authors have not indicates whihc statistical method they followed for p stimation in material and methods.
In the methods section, we have clarified which statistical methods were use
The conclusion is poor and it is not really supported by these finginds given the low size sample.
As suggested by the reviewers, we have modified the conclusions at the end of the article.
Two native speakers from the US and Canada made independent significant linguistic revisions of the paper.
Reviewer 2 Report
The manuscript submitted for publication to vaccines by Sypien and Zielonka is an interesting article investigating the knowledge, attitude, and awareness levels of parents regarding HPV vaccinations. This is an interesting and well organized paper that is extending potentially interesting and valuable public health value.
The reviewer would like to raise the following points:
1. How was the number of participants determined?
2. How do the HPV statistics in Poland compare to the rest of Europe and te Global ones?
3. Consider including the research question(s) and hypotheses at the end of the introduction section.
4. What were the inclusion and exclusion criteria for participating in the study?
5. The manuscript seems to be jumping from results section to conclusions without presenting a discussion section. Please consider structuring the manuscript with the sections delineated.
6. How do the findings of the authors compare to findings in terms of reasons and rationale towards the vaccine compared to other settings (countries)?
Author Response
The manuscript submitted for publication to vaccines by Sypien and Zielonka is an interesting article investigating the knowledge, attitude, and awareness levels of parents regarding HPV vaccinations. This is an interesting and well organized paper that is extending potentially interesting and valuable public health value.
Thank you for the positive evaluation of the work and valuable comments that we tried to take into account.
The reviewer would like to raise the following points:
- How was the number of participants determined?
387 parents were present in the hospital with their children during the study period September 20-24, 2021, and 302 agreed to complete the questionnaire
- How do the HPV statistics in Poland compare to the rest of Europe and te Global ones?
We do not have HPV statistics, but the statistics on cervical cancer indicate it indirectly. Cervical cancer incidence in Poland is higher than in other European countries. We have supplemented the introduction with this information.
- Consider including the research question(s) and hypotheses at the end of the introduction section.
We included the research question at the end of the introduction section.
- What were the inclusion and exclusion criteria for participating in the study?
We added the inclusion and exclusion criteria in the methods section.
- The manuscript seems to be jumping from results section to conclusions without presenting a discussion section. Please consider structuring the manuscript with the sections delineated.
This was an obvious editorial error. During the corrections, the word Discussion was accidentally deleted. Of course, we improved it.
- How do the findings of the authors compare to findings in terms of reasons and rationale towards the vaccine compared to other settings (countries)?
By discussing various problems, we compeer our results for research in different countries. Indirectly, such a comparison is contained in the text.
Reviewer 3 Report
Thank you for the opportunity to review this manuscript. Cervical cancer is a common cause of cancer in women worldwide Health literacy regarding HPV testing may influence the adoption of this behavior. The topic is very timely. Covid 19 pandemic has had a significant negative effect on oncology screenings, but cancer incidence has increased.
However, I have some suggestions to improve this research paper.
The introduction talks about determinants of organized cancer screening. In this context, I think it’s very important to introduce the Precede-proceed model. It assumes that health and health behaviors are influenced by multiple factors - epidemiological, socio-psychological, administrative, political, environmental – which must be considered and assessed for their modifiability to ensure effective interventions and it helps planning action to improve screening programs.
Here are some references “Saulle R, Sinopoli A, De Paula Baer A, Mannocci A, Marino M, De Belvis AG, Federici A, La Torre G. The PRECEDE-PROCEED model as a tool in Public Health screening: a systematic review. Clin Ter. 2020 Mar-Apr;171(2):e167-e177. doi: 10.7417/CT.2020.2208. PMID: 32141490”, “Cereda D, Federici A, Guarino A, Serantoni G; Gruppo PRECEDE-PROCEED, Coppola L, Lemma P, Rossi PG. Development and first application of an audit system for screening programs based on the PRECEDE-PROCEED model: an experience with breast cancer screening in the region of Lombardy (Italy). BMC Public Health. 2020 Nov 25;20(1):1778. doi: 10.1186/s12889-020-09842-8. PMID: 33238924; PMCID: PMC7687705”.
In addition, it’s important to highlight the correlation between health literacy and cancer screening with suitable references: “Baccolini V, Isonne C, Salerno C, Giffi M, Migliara G, Mazzalai E, Turatto F, Sinopoli A, Rosso A, De Vito C, Marzuillo C, Villari P. The association between adherence to cancer screening programs and health literacy: A systematic review and meta-analysis. Prev Med. 2022 Feb;155:106927. doi: 10.1016/j.ypmed.2021.106927. Epub 2021 Dec 23. PMID: 34954244”; “Oldach, B. R., & Katz, M. L. (2014). Health literacy and cancer screening: a systematic review. Patient education and counseling, 94(2), 149-157.”
Methods are well structured.
The dscussion is missing and the conclusion are unsatisfactory. They must be implemented.
Author Response
Thank you for the opportunity to review this manuscript. Cervical cancer is a common cause of cancer in women worldwide Health literacy regarding HPV testing may influence the adoption of this behavior. The topic is very timely. Covid 19 pandemic has had a significant negative effect on oncology screenings, but cancer incidence has increased.
Thank you for the positive evaluation of the work and valuable comments that we tried to take into account.
However, I have some suggestions to improve this research paper.
The introduction talks about determinants of organized cancer screening. In this context, I think it’s very important to introduce the Precede-proceed model. It assumes that health and health behaviors are influenced by multiple factors - epidemiological, socio-psychological, administrative, political, environmental – which must be considered and assessed for their modifiability to ensure effective interventions and it helps planning action to improve screening programs..
Thank you for this comment. We took advantage of it for the introduction
Here are some references “Saulle R, Sinopoli A, De Paula Baer A, Mannocci A, Marino M, De Belvis AG, Federici A, La Torre G. The PRECEDE-PROCEED model as a tool in Public Health screening: a systematic review. Clin Ter. 2020 Mar-Apr;171(2):e167-e177. doi: 10.7417/CT.2020.2208. PMID: 32141490”, “Cereda D, Federici A, Guarino A, Serantoni G; Gruppo PRECEDE-PROCEED, Coppola L, Lemma P, Rossi PG. Development and first application of an audit system for screening programs based on the PRECEDE-PROCEED model: an experience with breast cancer screening in the region of Lombardy (Italy). BMC Public Health. 2020 Nov 25;20(1):1778. doi: 10.1186/s12889-020-09842-8. PMID: 33238924; PMCID: PMC7687705”.
We added these articles to references.
In addition, it’s important to highlight the correlation between health literacy and cancer screening with suitable references: “Baccolini V, Isonne C, Salerno C, Giffi M, Migliara G, Mazzalai E, Turatto F, Sinopoli A, Rosso A, De Vito C, Marzuillo C, Villari P. The association between adherence to cancer screening programs and health literacy: A systematic review and meta-analysis. Prev Med. 2022 Feb;155:106927. doi: 10.1016/j.ypmed.2021.106927. Epub 2021 Dec 23. PMID: 34954244”; “Oldach, B. R., & Katz, M. L. (2014). Health literacy and cancer screening: a systematic review. Patient education and counseling, 94(2), 149-157.”
We also used this suggestion in the introduction and added the indicated article to the references.
Methods are well structured.
Thank you for this opinion
The dscussion is missing and the conclusion are unsatisfactory. They must be implemented.
This was an obvious editorial error. During the corrections, the word Discussion was accidentally deleted. Of course, we improved it. As suggested, we have changed the conclusions at the end of this paper.
Round 2
Reviewer 2 Report
The authors have made a reasonable effort in addressing reviewer's comments. Proofreading is suggested.
Reviewer 3 Report
I accept manuscript in the new form.